# MoVE: Synergistic Integration of Temporal Dependencies and Cross-Variable Experts for Efficient Multivariate Time Series Forecasting

## Abstract

Multivariate time-series forecasting presents the dual challenge of modeling intricate temporal dynamics and complex cross-variable dependencies. Prevailing approaches often prioritize one aspect at the expense of the other, leading to suboptimal performance. To address this limitation, we introduce MoVE, a novel framework that synergistically integrates temporal and cross-variable modeling within a unified architecture. MoVE employs two specialized experts. A Temporal Expert for capturing long-range dependencies and a lightweight Cross-Variable Expert for modeling robust cross-variable interactions. By decoupling these components within a Mixture-of-Experts framework and optimizing them collaboratively, MoVE dynamically adapts to diverse forecasting scenarios. Extensive experiments demonstrate that our framework achieves superior performance, establishing a new paradigm for effective multivariate timer series forecasting.

## 1 Introduction

Accurate Multivariate Time Series Forecasting (MTSF) is a critical component in numerous domains, including energy grid management, environmental monitoring, and financial market analysis. The core challenge lies in developing models that can simultaneously capture two distinct types of information: the intrinsic temporal patterns within each variable and the complex cross-variable dependencies that define their interrelationships.

Recent research has advanced along two primary trajectories. Indirect methods, such as SparseTSF Lin et al. (2025b), infer periodic patterns through latent representations, while direct techniques, like CycleNet Lin et al. (2024), explicitly encode periodicity with learnable vector. Concurrently, significant progress has been made in modeling both dependencies together in harmony, with studies like TQNet (Lin et al., 2025a) enhancing the representation of both dependencies with aforementioned vector in cross-attention. Despite these advancements, a significant gap remains: the lack of a unified, adaptive framework that explicitly and effectively collaborates powerful temporal modeling with robust cross-variable dependency learning.

To bridge this gap, we propose MoVE (Mixture of Various Experts), a novel MoE-based architecture designed to integrate these capabilities seamlessly and dynamically. MoVE decouples the modeling of temporal periodicity and variable dependencies into two dedicated expert modules. This design allows each component to be optimized for its specific task while a dynamic gating mechanism modulates their contributions based on the input context. This integrated yet decoupled approach facilitates efficient learning of complex inter-variable relationships without compromising the fidelity of temporal pattern extraction. Our contributions are threefold:

### 1.1 Contributions

In this work, the contributions are summarized as follow:

- **Mixture-of-Various-Experts (MoVE) Architecture**: We propose a novel framework that decouples temporal and cross-variable modeling through dynamic expert blending. MoVE

integrates CycleNet Lin et al. (2024) (a periodicity-focused temporal expert) via: a residual connection preserving temporal features, and act as a periodicity enhancement for the Recurrent Cross-Variable Transformer Encoder (RCVTE) through Periodicity-Aware Local Curriculum Learning (PALCL). The architecture employs four specialized pathways: two RCVTEs (for periodic and raw input streams) with their corresponding linear bypasses, dynamically blended by a linear gating network that weights contributions based on input context.

- **Recurrent Cross-Variable Transformer Encoder (RCVTE)**: A single-layer transformer encoder is introduced to recurrently processes both periodic and raw input streams to model robust channel interactions. RCVTE leverages Periodicity-Aware Local Curriculum Learning (PALCL) for stable training and a novel Robust Cross-Variable Attention (RCVA) mechanism in Equation 3 to prevent attention collapse, enabling effective inter-variable relationship capture with minimal computational overhead.

- **Empirical Validation**: MoVE's superior performance and structural advantages are analysed through comprehensive evaluations on standard long-term multivariate forecasting benchmarks (ETTh, ETTm, and Weather), including ablation studies of expert pathways and gating mechanisms.

## 2 RELATED WORK

The pursuit of accurate Multivariate Time Series Forecasting (MTSF) has catalyzed significant innovation in deep learning architectures, with a central point of divergence being the architectural treatment of inter-variable relationships. Contemporary research has coalesced around three distinct paradigms, each presenting a different philosophy for handling the complex interplay between temporal dynamics and cross-channel dependencies.

**Channel-Mixing Model** represent an early and intuitive strategy, embedding the entire multivariate vector at each time step into a unified latent representation. Seminal works in this category, such as Informer (Zhou et al., 2021) and Autoformer (Wu et al., 2021), leverage this shared embedding space to capture global temporal contexts through powerful attention mechanisms. This approach inherently suffers from homogenization of variable-specific characteristics, forcing the model to learn a one-size-fits-all representation that often blurs critical channel-specific patterns. Consequently, these complex models are frequently outperformed by simpler linear baselines on heterogeneous multivariate benchmarks (Zhou et al., 2021; Wu et al., 2021).

**Channel-Independent (CI) Model** emerged as a powerful counter-paradigm, advocating for processing each univariate series through shared model parameters with no explicit cross-channel communication. The remarkable success of this approach—evident in PatchTST (Nie et al., 2023), TimeMixer (Wang et al., 2024a), SparseTSF (Lin et al., 2025b), and CycleNet (Lin et al., 2024) is underpinned by robust avoidance of overfitting and significant computational efficiency. However, its core premise is also its greatest weakness: architecturally forbidding information exchange between variables renders CI models fundamentally incapable of leveraging rich interdependencies in real-world multivariate systems.

**Channel-Dependent (CD) Model** explicitly aim to model cross-variable interactions through attention-based architectures (Liu et al., 2024; Ilbert et al., 2024; Wang et al., 2024b) or MLP mixers (Wu et al., 2023; Chen et al., 2023). Although promising for learning variable interactions, many CD models neglect periodicity, a fundamental pillar of time series. Classical approaches like Seasonal-Trend Decomposition (STD) are embedded in Autoformer Wu et al. (2021) and FEDformer Zhou et al. (2023), but recent advances like CycleNet Lin et al. (2024) introduce learnable cycle vectors for explicit periodic pattern encoding. Despite these strides, a critical architectural gap remains: the dynamic, adaptive integration of periodic modeling with cross-variable dependency modeling. Most approaches handle these aspects in a static or sequential manner, failing to capture their deep intertwinement.

This inadequacy is starkly illustrated by TQNet Lin et al. (2025a), which forces synergy by using CycleNet's cycle vectors as fixed queries in cross-variable attention. This approach fails on two fronts: It exacerbates attention collapse (Ilbert et al., 2024), where transformer models rapidly converge to degenerate attention maps due to outlier sensitivity and stochastic noise, critically documented in

time series contexts (Huang et al., 2023); It employs a rigid integration strategy without adaptive modulation, preventing dynamic balancing between periodic and cross-variable features.

Our proposed MoVE (Mixture of Various Experts) framework transcends these limitations through a modularized expert-based approach. Instead of monolithic integration, MoVE decomposes MTSF into two core competencies: a temporal expert inspired by CycleNet Lin et al. (2024) and a Cross-Variable Expert. The latter is realized through our Recurrent Cross-Variable Transformer Encoder (RCVTE), which incorporates a stabilized attention mechanism (RCVA) to circumvent attention collapse (Ilbert et al., 2024; Huang et al., 2023). The innovation of MoVE lies in its dynamic gating network, which continuously and adaptively modulates expert contributions based on input context allowing seamless specialization on highly periodic signals (with weak inter-variable dependencies) or aperiodic series (with rich cross-channel correlations). This principled framework addresses the core architectural gap unmet by CI, CM, and CD paradigms, enabling robust unification of temporal and cross-variable learning.

## 3 METHODOLOGY

### 3.1 PROBLEM FORMULATION

In multivariate time series forecasting, the objective is to predict future values $P$ for each channel, represented as $Y_t \in \mathbb{R}^{C \times P}$. This prediction is based on historical time series data of length $L$ across $C$ channels, encapsulated in the input matrix $X_t \in \mathbb{R}^{C \times L}$. The goal is to train a predictive model $f_\omega : \mathbb{R}^{C \times L} \to \mathbb{R}^{C \times P}$, parameterized by $\omega$, to minimize the mean squared error (MSE) between the predicted and actual values.

### 3.2 ARCHITECTURAL OVERVIEW OF MOVE

The Mixture of Various Experts (MoVE) framework presents a novel architecture designed to synergistically model the two core components of multivariate time series: complex temporal dynamics and intricate cross-variable dependencies. Unlike rigid fusion techniques, MoVE employs a dynamic gating network to adaptively blend the contributions of four specialized expert pathways. A key innovation is the Periodicity-Aware Local Curriculum Learning (PALCL) strategy, which stabilizes training by prior learning from noise-free periodic signals before introducing the full non-stationary input. The overall architecture, depicted in Figure 1, is described in detail below.

As illustrated in the dataflow diagram, input sequences are first normalized via RevIN. The gating network then computes the blend weights to combine the outputs of the four distinct experts. The final prediction is generated through a weighted summation of expert output, augmented by the output of the temporal model (Lin et al., 2024) via a residual connection. This residual linkage ensures the preservation of strong temporal features and mitigates potential information loss through expert pathways.

#### 3.2.1 SPECIALIZED EXPERT PATHWAYS

MoVE integrates four expert pathways, organized into two complementary pairs processing different input streams:

- **Experts 1 & 2: Periodic Signal Processing** This pair processes the recurrent, noise-free periodic vector $\mathbf{V}_p$ derived from the temporal model (Lin et al., 2024). Expert 2 constitutes a *Periodic Cross-Variable Expert*, implemented as a Recurrent Cross-Variable Transformer Encoder (RCVTE), which captures stable periodic inter-channel relationships to enhance long-term dependency modeling. Expert 1 provides a simple linear projection that serves as a dynamic bypass pathway for $\mathbf{V}_p$, allowing the model to default to simpler transformations when complex interactions modeled by the RCVTE are unnecessary.

- **Experts 3 & 4: Original Input Processing.** This pair directly processes the original input. Expert 4 employs the same RCVTE architecture to learn robust cross-variable relationships from raw data, incorporating complex and non-stationary interaction while leveraging prior knowledge of periodic cross-variable relationships. Expert 3 provides a corresponding linear bypass, offering a simplified alternative pathway for processing flexibility.

To dynamically balance the contributions of the four experts, a lightweight gating network employs a softmax function. Apart from its basic functionality, This design strategically addresses two independent challenges in multivariate time series modeling.

First, it mitigates temporal distribution shift between training and test distributions, where severity varies across channels. As established in time series domain adaptation literature, such non-stationary significantly degrades model performance. The gating mechanism serves as an adaptive arbitrator between Channel Dependent (CD) and Channel Independent (CI) strategies, favoring CI for channels with severe shift and CD for stable channels. This hybrid approach aligns with findings from (Han et al., 2025), enabling channel-specific resilience to temporal characteristics.

Second, the gating network prevents model collapse—a degenerative condition where neural networks fail when processing overly homogeneous or synthetic data patterns. Drawing parallels to the recursive training collapse observed in generative AI (Shumailov et al., 2024), PALCL strategy faces similar risks when learned periodic components substantially overlap with raw inputs. This input similarity can create degenerate learning signals, analogous to training on recursively generated data. The gating mechanism counters this by maintaining expert diversity, selectively prioritizing or discounting pathways to preserve learning signal variety.

### 3.2.2 CROSS-VARIABLE EXPERT: RECURRENT CROSS-VARIABLE TRANSFORMER ENCODER (RCVTE)

The Recurrent Cross-Variable Transformer Encoder (RCVTE) module specifically addresses cross-variable dependency capture, overcoming two fundamental limitations of standard transformers in multivariate time series analysis. This is achieved through: (1) the integration of Periodicity-Aware Local Curriculum Learning (PALCL) to structure the transformer encoder's training progression, and (2) the replacement of conventional attention with Recurrent Cross-Variable Attention (RCVA) to enhance inter-channel relationship modeling.

**Periodicity-Aware Local Curriculum Learning (PALCL):**

We introduce a curriculum learning strategy (Bengio et al., 2009) that stabilizes training through a structured two-phase approach within each batch. In the first phase, RCVTE processes noise-free periodic recurrent vectors derived from CycleNet, establishing a simplified learning objective focused on fundamental periodic correlations. The second phase transitions to processing original data, enabling the model to integrate complex non-stationary patterns and residual components. This progressive curriculum ensures robust acquisition of core temporal interactions before introducing more challenging variations, as illustrated by the dotted line in Figure 1 denoting the recurrent encoder usage across phases.

**Robust Cross-Variable Attention (RCVA):** Standard softmax attention in Equation 1 demonstrates susceptibility to *attention collapse* (Ilbert et al., 2024), where weight concentration on outliers occurs due to exponential function sensitivity (Blanchard et al., 2020):

$$\text{Attention}(Q, K, V) = \text{Softmax}\left(\frac{QK^T}{\sqrt{d_k}}\right) V \tag{1}$$

Our proposed Robust Cross-Variable Attention (RCVA) mechanism addresses this limitation through numerically stable computation of the Log-Sum-Exp (LSE) function, defined for vector $x$ as:

$$\text{LSE}(x) = a + \log\left(\sum_{i=1}^{n} e^{x_i - a}\right), \quad \text{where } a = \max(x). \tag{2}$$

The RCVA mechanism applies GELU activation to stabilized logits:

$$\text{RCVA}(Q, K, V) = \text{Softmax}\left(\text{GELU}\left(A - \text{LSE}(A)\right)\right) V, \quad \text{where } A = \frac{QK^T}{\sqrt{d_k}} \tag{3}$$

This combined approach of LSE-based stabilization GELU Hendrycks & Gimpel (2017) to prevents attention collapse, promotes robust dynamic attention distribution, and suppresses outlier influence.

### 3.2.3 GATING NETWORK

An efficient gating network modulates expert contributions, inspired by Mixture-of-Experts (MoE) architectures Jacobs et al. (1991). The gating function $G(X_t)$ implements a linear projection followed by softmax activation:

$$G(X_t) = \text{Softmax}(W_g X_t) \tag{4}$$

where $W_g$ and $b_g$ represent learnable parameters. This generates adaptive mixing weights $[g_1, g_2, g_3, g_4]$ for the respective experts, producing the final integrated output:

$$\hat{Y}t = \sum_{i=1}^{4} g_i \cdot H_{\text{expert}_i}. \tag{5}$$

This architecture enables MoVE to dynamically prioritize different signal components (periodic, raw, simple, or cross-variable) based on a specific input context $X_t$.

### 3.2.4 REVERSE INSTANCE NORMALIZATION (REVIN)

To enhance robustness against non-stationary distribution shifts, we incorporate Reverse Instance Normalization Kim et al. (2021). This technique normalizes input sequences per channel by mean subtraction, with subsequent reversion of normalization during prediction output using original statistics. This process stabilizes training and improves generalization performance. (Standard deviation division is omitted in practice).

## 4 EXPERIMENTS

We conduct experiments on eight publicly available datasets of real-world multivariate time series that are widely recognized for long-term forecasting. These datasets include four ETT datasets (Zhou et al., 2021), Electricity, Weather, Traffic, and Solar are sourced from Autoformer Wu et al. (2021).

Table 1: Detailed Information of Datasets

| Category | ETTh1 & ETTh2 | ETTm1 & ETTm2 | Weather | Solar | Electricity | Traffic |
|---|---|---|---|---|---|---|
| **Timesteps** | 17,420 | 69,680 | 52,696 | 26,304 | 52,560 | 17,544 |
| **Channels** | 7 | 7 | 21 | 321 | 137 | 862 |
| **Frequency** | 1 hour | 15 mins | 10 mins | 1 hour | 10 mins | 1 hour |
| **Cyclic Patterns** | Daily | Daily | Daily | Weekly | Daily | Weekly |
| **Cycle Length** | 24 | 96 | 6 | 168 | 144 | 168 |

### 4.1 HYPERPARAMETER

Following the standardized experimental protocol established in previous work TQNet Lin et al. (2025a), all models are evaluated using a fixed look-back window of 96 time steps and random seed of 2024. This uniform setup ensures a fair and consistent comparison between methods, enabling us to assess MoVE's ability to capture both short-term and long-term patterns relative to existing approaches.

The hyperparameter configuration of MoVE is designed to balance model complexity and generalization by leveraging a scaling law that relates key hyperparameters to the number of variables in the dataset. This ensures that the model's capacity scales appropriately with the number of variables, mitigating overfitting or underfitting risks across diverse forecast horizons. See Table 2. For consistency, each dataset employs a uniform learning rate across all prediction horizons: most datasets use 0.0001, while Electricity and Traffic achieve optimal convergence with a higher learning rate of 0.0005. All configurations incorporate the StepLR (Paszke et al., 2019) learning rate scheduler to facilitate stable training. This tailored approach ensures efficient convergence while maintaining adaptability to varying data complexities.

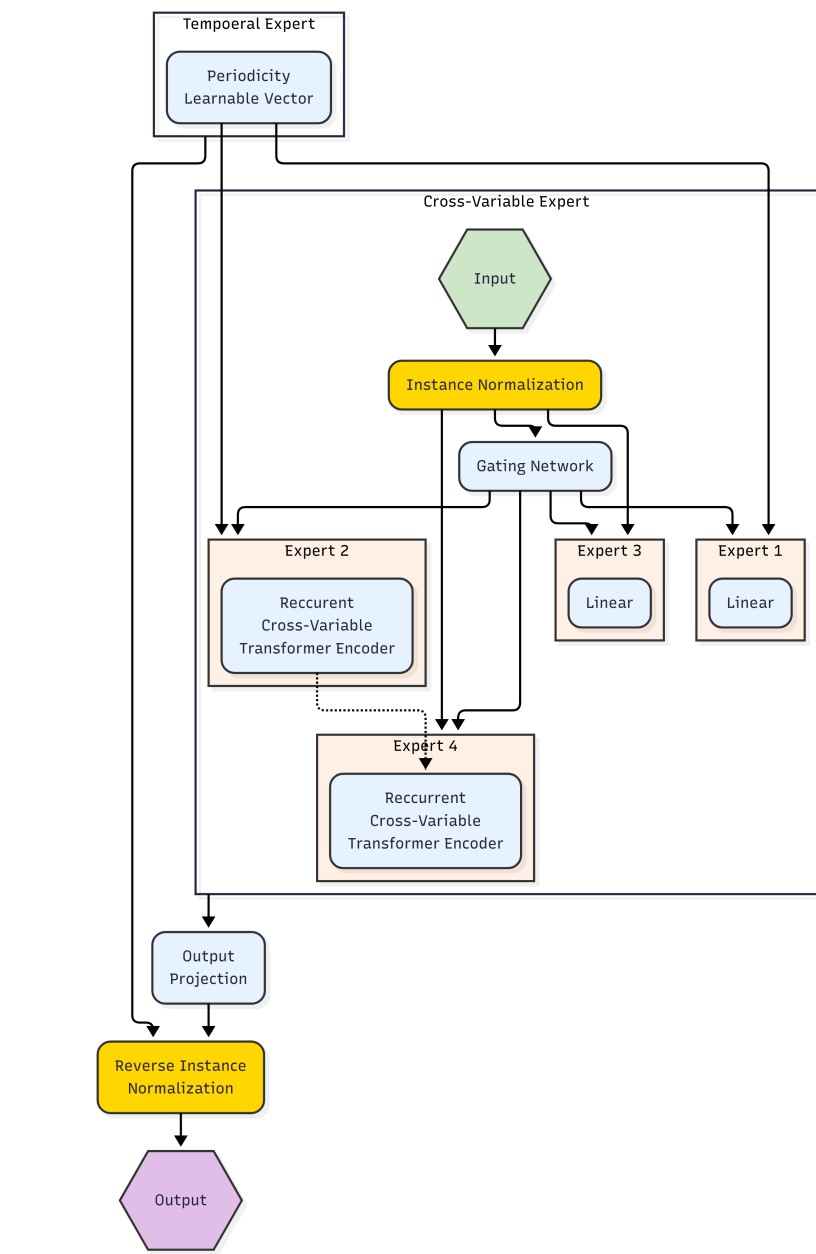

Figure 1: Overall structure of MoVE.

## 4.2 BASELINE

To rigorously evaluate the effectiveness of the proposed MoVE framework, we benchmark its performance against a comprehensive suite of advanced models representing the state-of-the-art in time series forecasting. Our baseline models include both recent and established methods: TQNet Lin et al. (2025a), TimeXer Wang et al. (2024b), CycleNet Lin et al. (2024), iTransformer Liu et al. (2024), MSGNet Cai et al. (2023), TimesNet Wu et al. (2023), PatchTST Nie et al. (2023), Crossformer Wang et al. (2023), DLinear Zeng et al. (2023).

Table 2: Classifications of Datasets with corresponding hyperparameters. The table demonstrates a clear positive relationship between number of variables and number of heads ($N$), as well as the dimension size for attention $D$ and for Feed-Forward Network $F$ to better capture the channel interaction (Vaswani et al., 2017).

| Dataset Type | Hyperparameters | Datasets |
|---|---|---|
| Low Variates - 7 | $N = 4, D = 32, F = 64$ | ETTH1, ETTH2, ETTM1, ETTM2 |
| Medium-Low Variates - 21 | $N = 8, D = 32, F = 128$ | Weather |
| Medium-High Variates - 137 | $N = 8, D = 128, F = 512$ | Solar |
| High Variates - 321 | $N = 16, D = 512, F = 1024$ | ECL |
| Ultra-High Variates - 862 | $N = 16, D = 1024, F = 2048$ | Traffic |

Table 3: Results for MTSF model across eight datasets by averaging performance among four prediction horizons $H \in \{96, 192, 336, 720\}$ with fixed lookback length of 96. Complete results are provided in Table 4. The best results are highlighted in **bold** and second best are underlined. Except for the results from MoVE, remaining results are sourced from TQNet Lin et al. (2025a).

| Dataset | MoVE | | TQNet | | TimeXer | | Cyclenet | | iTransformer | | MSGNET | | TimesNET | | PatchTST | | Crossformer | | DLinear | |
|---|---|---|---|---|---|---|---|---|---|---|---|---|---|---|---|---|---|---|---|---|
| Metric | MSE | MAE | MSE | MAE | MSE | MAE | MSE | MAE | MSE | MAE | MSE | MAE | MSE | MAE | MSE | MAE | MSE | MAE | MSE | MAE |
| ETTh1 | **0.417** | **0.429** | 0.441 | 0.434 | 0.437 | 0.437 | 0.457 | 0.441 | 0.454 | 0.448 | 0.453 | 0.453 | 0.458 | 0.450 | 0.469 | 0.455 | 0.529 | 0.522 | 0.456 | 0.452 |
| ETTh2 | 0.373 | 0.405 | 0.378 | 0.402 | **0.368** | **0.396** | 0.388 | 0.409 | 0.383 | 0.407 | 0.413 | 0.427 | 0.414 | 0.427 | 0.387 | 0.407 | 0.942 | 0.684 | 0.559 | 0.515 |
| ETTm1 | **0.376** | 0.402 | 0.377 | **0.393** | 0.382 | 0.397 | 0.379 | 0.396 | 0.407 | 0.410 | 0.400 | 0.412 | 0.400 | 0.406 | 0.387 | 0.400 | 0.513 | 0.495 | 0.403 | 0.407 |
| ETTm2 | 0.269 | 0.322 | 0.277 | 0.323 | 0.274 | 0.322 | **0.266** | **0.314** | 0.288 | 0.332 | 0.289 | 0.330 | 0.291 | 0.333 | 0.281 | 0.326 | 0.757 | 0.611 | 0.350 | 0.401 |
| Electricity | **0.158** | **0.256** | 0.164 | 0.259 | 0.171 | 0.270 | 0.168 | 0.259 | 0.178 | 0.270 | 0.194 | 0.301 | 0.193 | 0.295 | 0.205 | 0.290 | 0.244 | 0.334 | 0.212 | 0.300 |
| Solar | **0.191** | **0.249** | 0.198 | 0.256 | 0.237 | 0.302 | 0.210 | 0.261 | 0.233 | 0.262 | 0.263 | 0.292 | 0.301 | 0.319 | 0.270 | 0.307 | 0.641 | 0.639 | 0.330 | 0.401 |
| Traffic | 0.479 | 0.279 | 0.445 | **0.276** | 0.466 | 0.287 | 0.472 | 0.301 | **0.428** | 0.282 | 0.660 | 0.382 | 0.620 | 0.336 | 0.481 | 0.300 | 0.550 | 0.304 | 0.625 | 0.383 |
| Weather | 0.253 | 0.295 | 0.242 | 0.269 | **0.241** | 0.271 | 0.243 | 0.271 | 0.258 | 0.278 | 0.249 | 0.278 | 0.259 | 0.287 | 0.259 | 0.273 | 0.259 | 0.315 | 0.265 | 0.317 |

## 4.3 Environment

All experiments were carried out using PyTorch Paszke et al. (2019), with the Adam optimizer Kingma & Ba (2014), and executed on a single NVIDIA A10 GPU (24 GB memory).

## 4.4 Performance Analysis

We evaluated MoVE by comparing it with baseline models across eight real-world datasets. Lower values of Mean Squared Error (MSE) and Mean Absolute Error (MAE) indicate superior forecasting accuracy. Overall, MoVE achieves the best performance in 7 out of 16 tasks and secures second-best results in 5 of the rest takss. See Table 3. Notably, MoVE establishes new state-of-the-art (SOTA) benchmarks on the ETTH1, Electricity and Solar datasets to the best of our knowledge, highlighting its exceptional capability to model intertwinement between complex cross-variable interactions with temporal dependencies.

Despite relying on traditional self-attention, a mechanism which is often considered suboptimal. MoVE incorporates the PALCL module, which introduces a two-phase self-attention mechanism. This effectively mitigates the detrimental effects of noise disturbances (Lin et al., 2025a). Similarly, RCVTE adopts an approach inspired by iTransformer Liu et al. (2024), which inherently suffers from inadequate explicit temporal or periodicity modeling. However, this limitation is addressed through dynamic integration of learned periodic parameters (Lin et al., 2024). For CI-based approaches like CycleNet Lin et al. (2024) and TimeMixer Wang et al. (2024a) which lack explicit cross-variable modeling, RCVTE with PALCL manage to alleviates this deficiency.

To comprehensively understand how all components synergize and contribute to the framework's performance, we conduct an ablation study, detailed below.

Table 4: Full results provided for the performance comparison for all models in different datasets and prediction horizons $H \in \{96, 192, 336, 720\}$ with fixed look-back length of 96. The best results are highlighted in **bold** and second best results are underlined separately for MSE and MAE. Except the result from MoVE, rest of the result are source from TQNet (Lin et al., 2025a).

| Model | MoVE | | TQNet (2025b) | | TimeXer (2024c) | | CycleNet (2024b) | | iTransformer (2024c) | | MSGNet (2024) | | TimesNet (2023) | | PatchTST (2023) | | Crossformer (2023) | | DLinear (2023) | |
|---|---|---|---|---|---|---|---|---|---|---|---|---|---|---|---|---|---|---|---|---|
| Metric | MSE | MAE | MSE | MAE | MSE | MAE | MSE | MAE | MSE | MAE | MSE | MAE | MSE | MAE | MSE | MAE | MSE | MAE | MSE | MAE |
| ETTh1 96 | **0.368** | 0.394 | 0.371 | **0.393** | 0.382 | 0.403 | 0.375 | 0.395 | 0.386 | 0.405 | 0.390 | 0.411 | 0.384 | 0.402 | 0.414 | 0.419 | 0.423 | 0.448 | 0.386 | 0.400 |
| ETTh1 192 | **0.413** | **0.422** | 0.428 | 0.426 | 0.429 | 0.435 | 0.436 | 0.428 | 0.441 | 0.436 | 0.443 | 0.442 | 0.436 | 0.429 | 0.460 | 0.445 | 0.471 | 0.474 | 0.437 | 0.432 |
| ETTh1 336 | **0.448** | **0.442** | 0.476 | 0.446 | 0.468 | 0.448 | 0.496 | 0.455 | 0.487 | 0.458 | 0.482 | 0.469 | 0.491 | 0.469 | 0.501 | 0.466 | 0.570 | 0.546 | 0.481 | 0.459 |
| ETTh1 720 | **0.439** | **0.458** | 0.487 | 0.470 | 0.469 | 0.461 | 0.520 | 0.484 | 0.503 | 0.491 | 0.496 | 0.488 | 0.521 | 0.500 | 0.500 | 0.488 | 0.653 | 0.621 | 0.519 | 0.516 |
| ETTh2 96 | **0.285** | 0.346 | 0.295 | 0.343 | 0.286 | **0.338** | 0.298 | 0.344 | 0.297 | 0.349 | 0.329 | 0.371 | 0.340 | 0.374 | 0.302 | 0.348 | 0.745 | 0.584 | 0.333 | 0.387 |
| ETTh2 192 | 0.366 | 0.392 | 0.367 | 0.393 | **0.363** | **0.389** | 0.372 | 0.396 | 0.380 | 0.400 | 0.402 | 0.414 | 0.402 | 0.414 | 0.388 | 0.400 | 0.877 | 0.656 | 0.477 | 0.476 |
| ETTh2 336 | 0.415 | 0.431 | 0.417 | 0.427 | **0.414** | **0.423** | 0.431 | 0.439 | 0.428 | 0.432 | 0.440 | 0.445 | 0.452 | 0.452 | 0.426 | 0.433 | 1.043 | 0.731 | 0.594 | 0.541 |
| ETTh2 720 | 0.428 | 0.450 | 0.433 | 0.446 | **0.408** | **0.432** | 0.450 | 0.458 | 0.427 | 0.445 | 0.480 | 0.477 | 0.462 | 0.468 | 0.431 | 0.446 | 1.104 | 0.763 | 0.831 | 0.657 |
| ETTm1 96 | 0.324 | 0.372 | **0.311** | **0.353** | 0.318 | 0.356 | 0.319 | 0.360 | 0.334 | 0.368 | 0.319 | 0.366 | 0.338 | 0.375 | 0.329 | 0.367 | 0.404 | 0.426 | 0.345 | 0.372 |
| ETTm1 192 | 0.363 | 0.393 | **0.356** | **0.378** | 0.362 | 0.383 | 0.360 | 0.381 | 0.377 | 0.391 | 0.377 | 0.397 | 0.374 | 0.387 | 0.367 | 0.385 | 0.450 | 0.451 | 0.380 | 0.389 |
| ETTm1 336 | **0.386** | 0.408 | 0.390 | **0.401** | 0.395 | 0.407 | 0.389 | 0.403 | 0.426 | 0.420 | 0.417 | 0.422 | 0.410 | 0.411 | 0.399 | 0.410 | 0.532 | 0.515 | 0.413 | 0.413 |
| ETTm1 720 | **0.431** | **0.434** | 0.452 | 0.440 | 0.452 | 0.441 | 0.447 | 0.441 | 0.491 | 0.459 | 0.487 | 0.463 | 0.478 | 0.450 | 0.454 | 0.439 | 0.666 | 0.589 | 0.474 | 0.453 |
| ETTm2 96 | 0.167 | 0.258 | 0.173 | 0.256 | 0.171 | 0.256 | **0.163** | **0.246** | 0.180 | 0.264 | 0.182 | 0.266 | 0.187 | 0.267 | 0.175 | 0.259 | 0.287 | 0.366 | 0.193 | 0.292 |
| ETTm2 192 | 0.231 | 0.299 | 0.238 | 0.298 | 0.237 | 0.299 | **0.229** | **0.290** | 0.250 | 0.309 | 0.248 | 0.306 | 0.249 | 0.309 | 0.241 | 0.302 | 0.414 | 0.492 | 0.284 | 0.362 |
| ETTm2 336 | 0.290 | 0.336 | 0.301 | 0.340 | 0.296 | 0.338 | **0.284** | **0.327** | 0.312 | 0.346 | 0.321 | 0.351 | 0.305 | 0.343 | 0.341 | 0.343 | 0.597 | 0.542 | 0.369 | 0.427 |
| ETTm2 720 | **0.388** | 0.395 | 0.397 | 0.396 | 0.392 | 0.394 | 0.389 | **0.391** | 0.412 | 0.407 | 0.414 | 0.404 | 0.408 | 0.403 | 0.402 | 0.400 | 1.730 | 1.042 | 0.554 | 0.522 |
| ECL 96 | **0.132** | 0.230 | 0.134 | **0.229** | 0.140 | 0.242 | 0.136 | **0.229** | 0.148 | 0.244 | 0.165 | 0.274 | 0.168 | 0.272 | 0.181 | 0.270 | 0.219 | 0.314 | 0.197 | 0.282 |
| ECL 192 | **0.148** | 0.245 | 0.154 | 0.247 | 0.157 | 0.256 | 0.152 | **0.244** | 0.162 | 0.253 | 0.185 | 0.292 | 0.184 | 0.289 | 0.188 | 0.274 | 0.231 | 0.322 | 0.196 | 0.285 |
| ECL 336 | **0.161** | **0.261** | 0.169 | 0.264 | 0.176 | 0.275 | 0.170 | 0.264 | 0.178 | 0.269 | 0.197 | 0.304 | 0.198 | 0.300 | 0.204 | 0.293 | 0.246 | 0.337 | 0.209 | 0.301 |
| ECL 720 | **0.190** | **0.290** | 0.201 | 0.294 | 0.211 | 0.306 | 0.212 | 0.299 | 0.225 | 0.317 | 0.231 | 0.332 | 0.220 | 0.320 | 0.246 | 0.324 | 0.280 | 0.363 | 0.245 | 0.333 |
| Solar 96 | **0.169** | 0.237 | 0.173 | **0.233** | 0.215 | 0.295 | 0.190 | 0.247 | 0.203 | 0.237 | 0.210 | 0.246 | 0.250 | 0.292 | 0.234 | 0.286 | 0.310 | 0.331 | 0.290 | 0.378 |
| Solar 192 | **0.192** | **0.249** | 0.199 | 0.257 | 0.236 | 0.301 | 0.210 | 0.266 | 0.210 | 0.266 | 0.265 | 0.290 | 0.296 | 0.318 | 0.267 | 0.310 | 0.734 | 0.725 | 0.320 | 0.398 |
| Solar 336 | **0.198** | **0.259** | 0.211 | 0.263 | 0.252 | 0.307 | 0.217 | 0.266 | 0.248 | 0.273 | 0.294 | 0.318 | 0.319 | 0.330 | 0.290 | 0.315 | 0.750 | 0.735 | 0.353 | 0.415 |
| Solar 720 | **0.205** | **0.257** | 0.209 | 0.270 | 0.244 | 0.305 | 0.223 | 0.266 | 0.285 | 0.315 | 0.285 | 0.315 | 0.338 | 0.337 | 0.289 | 0.317 | 0.769 | 0.765 | 0.356 | 0.413 |
| Traffic 96 | 0.459 | 0.266 | 0.413 | **0.261** | 0.428 | 0.271 | 0.458 | 0.296 | **0.395** | 0.268 | 0.608 | 0.349 | 0.593 | 0.321 | 0.462 | 0.290 | 0.522 | 0.296 | 0.650 | 0.396 |
| Traffic 192 | 0.473 | 0.272 | 0.432 | **0.271** | 0.448 | 0.282 | 0.457 | 0.294 | **0.417** | 0.276 | 0.617 | 0.336 | 0.629 | 0.336 | 0.530 | 0.293 | 0.598 | 0.305 | 0.605 | 0.373 |
| Traffic 336 | 0.479 | 0.282 | 0.450 | **0.277** | 0.473 | 0.289 | 0.470 | 0.299 | **0.433** | 0.283 | 0.669 | 0.388 | 0.629 | 0.336 | 0.482 | 0.300 | 0.558 | 0.305 | 0.605 | 0.373 |
| Traffic 720 | 0.505 | **0.295** | 0.486 | **0.295** | 0.516 | 0.307 | 0.502 | 0.314 | **0.467** | 0.302 | 0.729 | 0.420 | 0.640 | 0.350 | 0.514 | 0.320 | 0.589 | 0.328 | 0.645 | 0.394 |
| Weather 96 | 0.164 | 0.224 | **0.157** | **0.200** | 0.157 | 0.205 | 0.158 | 0.203 | 0.174 | 0.214 | 0.163 | 0.212 | 0.172 | 0.220 | 0.177 | 0.210 | 0.158 | 0.230 | 0.196 | 0.255 |
| Weather 192 | 0.215 | 0.267 | 0.206 | **0.245** | **0.204** | 0.247 | 0.207 | 0.247 | 0.221 | 0.254 | 0.211 | 0.254 | 0.219 | 0.261 | 0.225 | 0.250 | 0.206 | 0.277 | 0.237 | 0.296 |
| Weather 336 | 0.276 | 0.317 | 0.262 | **0.287** | **0.261** | 0.290 | 0.262 | 0.289 | 0.278 | 0.296 | 0.273 | 0.299 | 0.280 | 0.306 | 0.278 | 0.290 | 0.272 | 0.335 | 0.283 | 0.335 |
| Weather 720 | 0.357 | 0.371 | 0.344 | 0.342 | **0.340** | 0.341 | 0.344 | 0.344 | 0.358 | 0.349 | 0.351 | 0.348 | 0.365 | 0.359 | 0.354 | **0.340** | 0.398 | 0.418 | 0.345 | 0.381 |

Table 5: Ablation study of MoVE components across six datasets ($H = 720$, lookback=96). w/o Both: only PALCL strategy used

| Dataset | MoVE | | w/o RCVA | | w/o Gating | | w/o RCVA & Gating | | w/o PALCL | |
|---|---|---|---|---|---|---|---|---|---|---|
| | MSE | MAE | MSE | MAE | MSE | MAE | MSE | MAE | MSE | MAE |
| ETTh1 | **0.439** | **0.458** | 0.457 | 0.466 | 0.457 | 0.466 | 0.464 | 0.468 | 0.455 | 0.465 |
| ETTh2 | **0.428** | **0.450** | 0.428 | 0.451 | 0.432 | 0.455 | 0.431 | 0.454 | 0.432 | 0.455 |
| ETTm1 | 0.431 | 0.434 | 0.438 | 0.440 | **0.430** | **0.431** | 0.441 | 0.439 | 0.440 | 0.438 |
| ETTm2 | **0.388** | **0.395** | 0.393 | 0.396 | 0.388 | 0.396 | 0.388 | 0.395 | 0.390 | 0.395 |
| ECL | **0.190** | **0.290** | 0.203 | 0.300 | 0.202 | 0.300 | 0.208 | 0.304 | 0.195 | 0.294 |
| Solar | **0.205** | **0.257** | 0.225 | 0.271 | 0.206 | 0.260 | 0.207 | 0.265 | 0.207 | 0.265 |

## 4.5 Ablation Study and Further Analysis

### 4.5.1 Ablation Study of MoVE

To evaluate the contribution of the core components in MoVE, which dynamically integrates temporal and cross-variable modeling, we conduct an ablation study using the following variants:

1. **Without Gating Mechanism**: The gating mechanism, which serves as a dynamic residual connection to mitigate information loss in complex pathways, is replaced with simple residual connection He et al. (2016).

2. **Cross-Variable Standard Transformer**: The RCVA module is replaced with a standard multi-head self-attention mechanism Vaswani et al. (2017).

3. **Removal of Gating Mechanism and RCVA**: Both gating mechanism and RCVA module are disabled, leaving only the PALCL module active.

4. **Removal of PALCL**: PALCL is removed as not recurrent application of the transformer to perdiodic component and raw input

The results of the ablation study, summarized in Table 5, elucidate the individual and synergistic contributions of each component to MoVE's performance. The model achieves the best or highly competitive results across all datasets, with the relative importance of each module varying according to dataset characteristics.

The removal of the RCVA module induces the most pronounced performance drop, particularly on noisy ETT-series datasets, underscoring its critical role in capturing robust cross-variable dependencies. In contrast, the absence of the gating mechanism leads to a more subtle but consistent degradation, with significant declines observed only in the ETTh1 and ECL tasks. Similarly, ablating either the PALCL or RECVT components results in substantial performance deterioration on the ETTh1, ETTM1, and ECL datasets.

Notably, the full MoVE model demonstrates superior performance on complex datasets like ECL and Solar, indicating that the synergistic interplay of temporal modeling and cross-variable relationships is essential in these contexts. Interestingly, on Solar and ETTm2, the variant without both RCVA and gating surpasses the full model in certain metrics, suggesting that simpler interactions may suffice for specific data characteristics. This observation further validates MoVE's inherent adaptability in balancing component contributions to avoid superfluous complexity. Finally, the strong standalone performance of the PALCL baseline confirms its effectiveness as a robust foundational strategy for cross-variable modeling.

### 4.5.2 HYPERPARAMETER ANALYSIS

As shown in Table 2, we observe a positive correlation between the number of variables in a dataset and the optimal model hyperparameters. Specifically, both the model dimension $D$ and feedforward capacity $F$ scale proportionally with variable count, while maintaining parameter efficiency. This trend aligns with transformer architecture principles, where increased model capacity is required to capture complex cross-variable interactions effectively (Vaswani et al., 2017).

## 5 LIMITATIONS

Although the proposed framework effectively unifies temporal and cross-variable experts for time series forecasting, it has several limitations:

1. Similar to CycleNet Lin et al. (2024), which relies on periodicity, the PALCL module might underperform when periodic patterns are weak or obscure.

2. While MoVE dynamically mitigates the negative effects of forced cross-variable modeling, residual impacts might persist. Further research is needed to enhance the context-aware modulation of component contributions.

3. The model currently captures only a single periodicity, which may be insufficient for real-world data exhibiting multiple overlapping cycles (e.g., daily, weekly, and monthly periodicity).

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

# 6 APPENDIX

## 6.1 USE OF LARGE LANGUAGE MODEL

In the preparation of this work, we used Deepseek to polish the grammar and improve the clarity of writing in all sections. After using this tool, we reviewed and edited the content as needed and take full responsibility for the content of the publication, including any inaccuracies or errors that may remain.

## 6.2 REPRODUCIBILITY STATEMENT

To ensure reproducibility, we provide all codes, experimental configurations, and required items in an anonymous url: `https://anonymous.4open.science/r/MoVE-4DBC/`. Detailed protocols, dataset descriptions, hyperparameter settings, and environment specifications are available in Section 4. A shell script (`Linear-I.sh`) is included to facilitate the end-to-end reproduction of all experiments, including data preprocessing, model training, and evaluation. The repository also contains a `README.md` file with setup instructions, along with exact dependency specifications to ensure consistent results.

## 6.3 ETHICS STATEMENT

All datasets used in this study are publicly available and appropriately cited. We have taken care to ensure that our use of data complies with the terms of their respective licenses. This work is the original intellectual contribution of the authors. All external sources, including code, methodologies, and ideas, have been properly acknowledged and cited.

## 6.4 EXPERT ACTIVATION ANALYSIS

To investigate the specialization and functional roles of different experts within the MoVE framework, we conducted activation analysis using systematically generated artificial sequences. The model, trained on the ETTh1 dataset, was evaluated with two distinct input types: perfectly periodic sequences with 24-hour periodicity and complex aperiodic sequences with multiple frequency components.

The experimental setup, implemented through our gate inspection framework, generates controlled synthetic data while preserving the statistical properties of the original training distribution. Periodic sequences were constructed as synchronized sinusoidal waves with channel-varying amplitudes, while aperiodic sequences combined multiple base frequencies with randomized phases to simulate complex temporal patterns.

As shown in Table 6, the analysis reveals clear functional specialization among experts. Expert 1 demonstrates predominant activation for periodic sequences (mean gate weight: 0.68), indicating its specialization in modeling stable cross-variable relationships with strong temporal regularity. Conversely, Experts 3 and 4 show significantly higher activation for aperiodic sequences (combined mean weight: 0.72), suggesting their role in handling irregular patterns requiring minimal periodic cross-variable modeling.

This activation differentiation provides empirical evidence for the model's capacity to automatically route inputs through specialized processing pathways based on temporal characteristics. The gating mechanism successfully identifies periodic dominance to leverage Expert 1's periodic relationship modeling, while delegating complex aperiodic patterns to Experts 3 and 4's complementary processing strategies.

## 6.5 EFFECTIVENESS ANALYSIS

As evidenced in Table 7, the model maintains computational efficiency across diverse datasets despite its architectural complexity. The training time remains competitive due to the recurrent design that effectively functions as a two-stage transformer with parameter sharing, rather than two independent models. This architectural approach positions the model's computational profile between that of simpler and more complex alternatives.

Drawing from the TQNet study's layer-wise analysis (Lin et al., 2025a), where TQNet employs a single transformer layer and iTransformer utilizes three layers (Liu et al., 2024), our model's two-stage recurrent architecture naturally occupies an intermediate position in the computational spectrum. This parameter-sharing mechanism, combined with the parallelization capabilities of attention and Mixture-of-Experts (MoE) components, ensures that the additional computational overhead remains modest relative to the performance gains achieved.

Table 6: Comparison of gating values for MoVE with different types of input sequences

| Gating values | Expert 1 | Expert 2 | Expert 3 | Expert 4 |
|---|---|---|---|---|
| Periodicity-24 | 0.320 | 0.241 | 0.205 | 0.234 |
| Random | 0.173 | 0.218 | 0.302 | 0.307 |

Table 7: Average parameter counts and FLOPS of MoVE across eight datasets over four prediction horizons $H \in \{96, 192, 336, 720\}$.

|  | ETTh1 | ETTh2 | ETTM1 | ETTM2 | ECL | SOLAR | Traffic | Weather | Average |
|---|---|---|---|---|---|---|---|---|---|
| **Parameters** | 33.253K | 33.253K | 33.379K | 33.379K | 2526.077K | 310.885K | 9258.277K | 44.947K | 1534.181K |
| **FLOPS** | 0.287M | 0.287M | 0.287M | 0.287M | 1477.5M | 68.392M | 15165M | 1.407M | 2089.181M |

Table 8: Comparison of overall parameter counts of MTSF models across eight datasets. Except for the results from MoVE, remaining results are sourced from TQNet Lin et al. (2025a).

|  | MoVE | TQNet | CycleNet | iTransformer | TimeXer | PatchTST | DLinear |
|---|---|---|---|---|---|---|---|
| **Parameters** | ~1.5M | ~1M | ~300K | ~3M | ~10M | ~10M | ~137K |

