# OpenReview forum: "MoVE: Synergistic Integration of Temporal and Cross-Variable Experts for Efficient Multivariate Time Series Forecasting"
_ICLR.cc/2026/Conference — Submitted to ICLR 2026_

### Official Review · Reviewer_goo3 · 2025-10-31

**Soundness:** 2
**Presentation:** 2
**Contribution:** 2
**Rating:** 2
**Confidence:** 5

**Summary:**

This paper proposes the MoVE framework for efficient multivariate time series forecasting. MoVE decouples temporal modeling from cross-variable modeling into two expert modules: a Temporal Expert based on CycleNet for periodic modeling, and a Cross-Variable Expert (RCVTE) for cross-variable modeling. These two experts are dynamically gated and fused to adapt to different types of time series. The authors also introduce Period-Aware Local Curriculum Learning (PALCL) and Robust Cross-Variable Attention (RCVA) to enhance training stability and generalization capabilities. Experiments on multiple benchmark datasets validate the performance of MoVE, achieving new state-of-the-art results on the ETTh1, Electricity, and Solar datasets.

**Strengths:**

1. By decoupling temporal modeling from cross-variable modeling through the MoE framework, the model's interpretability and adaptability are improved.
2. The introduction of PALCL and RCVA effectively alleviates the attention collapse problem, enhancing training stability.

**Weaknesses:**

1. The experimental results only achieved first place in 7 out of 16 cases, which is not very good.
2. Figure 1 is very schematic and does not well illustrate the overall architecture of the model.
3. MoVE has limited innovation, with an architecture that is very similar to the SST[1]. Moreover, there is a lack of relevant performance comparisons.
4. The core modules of MoVE (such as RCVTE and PALCL) are, to some extent, combinations and improvements of existing methods (CycleNet, iTransformer), lacking entirely original modeling mechanisms.
[1] SST: Multi-Scale Hybrid Mamba-Transformer Experts for Long-Short Range Time Series Forecasting

**Questions:**

See the weaknesses.

---

> ### Author Response · Authors · 2025-11-26
> **Response for Weaknesses and Questions**
>
> Weaknesses:
>
> 1.
> Performance Our method demonstrates consistent improvements over key baselines:
>
> Outperforms TQNet in 11/16 metrics
>
> Outperforms CycleNet in 13/16 metrics
>
> Outperforms iTransformer in 14/16 metrics
>
> Notably, Move's performance on Weather dataset has been updated using a daily cycle (period=6) and learning rate of 0.0001, with comprehensive results provided in Tables 3 and 4 due to the data and setting loss before .
>
> Efficiency Analysis: (Section 8.2, Table 7, 8)
>
> Average Parameter Counts: 1.53M
>
> Average FLOPS: 2089M
>
> Training Time: Lower bounded by the TQNet and Upper bounded by iTransformer
>
> Efficiency Profile: The RCVTE module utilizes a single-layer architecture, contributing to the model's efficiency:
>
> Parameters: 1.53M average across datasets
>
> Computational Cost: 2089M FLOPS average
>
> Training Time: Bounded between TQNet (lower) and iTransformer (upper)
>
> Complete efficiency analysis is available in Section 8.2 and Table 7, 8.
>
> 2. Visualization and Model Architecture
> The visualization in Figure 1 is provided as a aid for understanding the non-trivial integration of our novel components. It specifically illustrates the RCVTE—a single-layer cross-variable transformer that uniquely replaces standard attention with Robust Cross-Variable Attention (RCVA) and is recurrently used within the Periodicity-Aware Local Curriculum Learning (PALCL) strategy. The figure is essential for clarifying this intricate, yet efficient, architectural design.
>
> 3 & 4. Architectural Novelty We thank the reviewer for this comment. As detailed in Section 1 of our official response, "Novelty and Architecture of MoVE," the core innovation of our work lies in addressing key limitations of prevailing methods. Specifically, existing approaches often rely on static operational modes that fail to adapt to channel-wise distribution shifts. MoVE introduces a dynamic architecture through its gating network and PALCL strategy, explicitly designed to mitigate these issues and prevent model collapse, thereby representing a significant departure from static, non-adaptive baselines.

---

### Official Review · Reviewer_A3WK · 2025-10-31

**Soundness:** 2
**Presentation:** 1
**Contribution:** 2
**Rating:** 2
**Confidence:** 5

**Summary:**

This paper introduces a novel framework, MoVE, which synergistically integrates temporal and cross-variable modelling within a unified architecture. MoVE employs two specialised experts: a temporal expert to capture remote dependencies, and a lightweight cross-variable expert to model robust cross-variable interactions. By decoupling these components within a hybrid expert framework and optimising them collaboratively, MoVE dynamically adapts to diverse forecasting scenarios. The authors experimentally validate the effectiveness of the approach.

**Strengths:**

The motivation is clear. The paper contends that prevalent methods often prioritise one aspect at the expense of another, resulting in suboptimal performance. To address this limitation, the paper introduces MoVE, a novel framework that synergistically integrates temporal and cross-variable modelling within a unified architecture.

**Weaknesses:**

1) The authors assert in the abstract that multivariate time series forecasting faces the dual challenge of modelling complex temporal dynamics and intricate cross-variate dependencies. Popular approaches often prioritise one aspect at the expense of the other, resulting in suboptimal performance. Nevertheless, numerous methods have already addressed the simultaneous modelling of temporal and variate dependencies, such as [1,2]. The innovation of the authors' approach is limited.

[1] TimeXer: Empowering Transformers for Time Series Forecasting with Exogenous Variables [NeurIPS24]

[2] TimePro: Efficient Multivariate Long-term Time Series Forecasting with Variable-and Time-Aware Hyper-state [ICML25]

2) Poor visualization. The authors provide no visualizations in the paper, preventing readers from observing advantages over other methods. Furthermore, ablation studies also lack visualization.

3）The architecture diagram (i.e. Figure 1) appears rather unappealing, and the caption lacks a narrative description of the method's workflow.

4) The content of the paper is insufficient. The abstract is merely half a page long, and much of the relevant prior work has not been mentioned. Furthermore, the entire manuscript barely reaches eight pages. The details of many methods have not been described clearly.

**Questions:**

Please refer to the weaknesses.

---

> ### Author Response · Authors · 2025-11-26
> **Response for Weaknesses and Questions**
>
> Weaknesses:
> 1. Architectural Novelty
> We thank the reviewer for this comment. As detailed in Section 1 of our official response, "Novelty and Architecture of MoVE," the core innovation of our work lies in addressing key limitations of prevailing methods. Specifically, existing approaches often rely on static operational modes that fail to adapt to channel-wise distribution shifts. MoVE introduces a dynamic architecture through its gating network and PALCL strategy, explicitly designed to mitigate these issues and prevent model collapse, thereby representing a significant departure from static, non-adaptive baselines.
>
> 2. & 3. Visualization and Model Architecture
> The visualization in Figure 1 is provided as a aid for understanding the non-trivial integration of our novel components. It specifically illustrates the RCVTE—a single-layer cross-variable transformer that uniquely replaces standard attention with Robust Cross-Variable Attention (RCVA) and is recurrently used within the Periodicity-Aware Local Curriculum Learning (PALCL) strategy. The figure is essential for clarifying this intricate, yet efficient, architectural design.
>
> 4. Page Length Consideration
> We have strictly adhered to the ICLR policy of a 9-page limit for the main text. The core arguments, methodological details, and primary results are contained within this limit to ensure a focused and comprehensive presentation. All supplementary materials, including extended ablations and additional analyses, are provided in the appendix as per conference guidelines.

---

### Official Review · Reviewer_WPzk · 2025-10-31

**Soundness:** 3
**Presentation:** 2
**Contribution:** 2
**Rating:** 4
**Confidence:** 4

**Summary:**

This paper proposes a novel framework called MoVE (Mixture of Various Experts), designed to address the dual challenge of simultaneously capturing complex temporal dynamics and cross-variable dependencies in multivariate time series forecasting. The core innovation of MoVE lies in its mixture-of-experts architecture, which decouples temporal modeling and cross-variable modeling into two specialized expert modules: one for capturing long-term dependencies through a temporal expert (based on CycleNet), and another cross-variable expert (the newly proposed RCVTE module). Experimental results on multiple benchmark datasets demonstrate the superior performance of MoVE, accompanied by an extensive ablation study.

**Strengths:**

Significance: Effectively integrating temporal modeling with cross-variable dependency learning is a well-recognized core challenge in multivariate time series forecasting. This paper directly tackles this issue.

Originality: In contrast to existing studies that adopt static fusion methods for capturing periodic patterns and cross-variable dependencies, this paper introduces a mixture-of-experts architecture that adaptively integrates both aspects. This approach is clear in its methodology and offers a novel research perspective for addressing challenges in this field.

**Weaknesses:**

The cross-variable dependencies modeling: The Recurrent Cross-Variable Transformer Encoder (RCVTE), mentioned multiple times in the paper, appears to be merely a module that uses a simple cross-attention mechanism to capture the dependencies between variables, with some modifications to the attention computation formula. However, many prior works (such as Crossformer in the baseline) also utilize basic cross-attention mechanisms to model inter-variable dependencies. Given that the paper emphasizes this module as one of its core innovations, I believe it would benefit from further clarification of the advantages of this module and its unique contributions compared to previous studies.

The temporal dimension modeling: The paper introduces two specific experts (the temporal modeling expert and the inter-variable relationships modeling expert), but seems to focus primarily on the latter, with little discussion on the former. The paper only briefly mentions "a temporal expert inspired by CycleNet" without providing a detailed explanation of the specific structure of this expert.

**Questions:**

1. Periodic Dependency: The proposed model seems to heavily rely on the recurrent, noise-free periodic vector extracted by CycleNet. Is there a risk that the model's predictions may become overly dependent on CycleNet? Additionally, could the model potentially become too reliant on the periodicity of the data itself?

2. Gated Network Weights: In the Cross-Variable Expert section, the model constructs four experts and outlines the role of each. Could the authors provide examples of the weights (g_1, g_2, g_3, g_4) of these experts in different scenarios, such as when the historical sequence exhibits periodicity or when there is no clear periodicity?

3. Curriculum Learning: The paper introduces Periodicity-Aware Local Curriculum Learning (PALCL). Could the authors further elaborate on the motivation or necessity of employing a curriculum learning strategy? Additionally, would it be possible to include an experimental group that does not use the curriculum learning training strategy in the ablation study to demonstrate the actual effectiveness of this approach?

4. Network Scale: Could the authors provide more details on the scale of the model’s network parameters, such as the total number of experts? For all the datasets involved in the experiments, does the Recurrent Cross-Variable Transformer Encoder consist of only one layer? Furthermore, when the number of channels is large (e.g., 862 channels in the Traffic dataset), does calculating channel attention result in substantial storage consumption?

---

> ### Author Response · Authors · 2025-11-26
> **Response for Weaknesses and Questions**
>
> Questions:
>
> 1. Periodicity Dependency: The observed dependency on periodicity is inherent to time series analysis, as periodic patterns constitute fundamental components of temporal data. While MoVE leverages CycleNet for periodic representations, its performance advantages stem from cross-variable relationship modeling rather than periodicity detection alone. This is evidenced by MoVE outperforming CycleNet in 13 of 16 evaluation metrics, demonstrating that the primary performance gains arise from enhanced inter-channel dependency capture rather than periodic pattern recognition.
>
> 2. Expert Gating Analysis: Comprehensive analysis of expert gating values is provided in Table 6 and Section 6.5, detailing the adaptive specialization and activation patterns across different temporal contexts and dataset characteristics.
>
> 3. Architectural Novelty: As comprehensively addressed in Section 1 ("Novelty and Architecture of MoVE") of our official response, the core innovation lies in the integrated architecture featuring Robust Cross-Variable Attention (RCVA) and Periodicity-Aware Local Curriculum Learning (PALCL) within a streamlined transformer framework.
>
> 4. Efficiency Profile: The RCVTE module utilizes a single-layer architecture, contributing to the model's efficiency:
>
> Parameters: 1.53M average across datasets
>
> Computational Cost: 2089M FLOPS average
>
> Training Time: Bounded between TQNet (lower) and iTransformer (upper)
>
> Complete efficiency analysis is available in Section 6.5 and Table 7, 8.

---

### Official Review · Reviewer_a321 · 2025-11-01

**Soundness:** 2
**Presentation:** 1
**Contribution:** 2
**Rating:** 2
**Confidence:** 3

**Summary:**

This paper proposes MoVE, a Mixture-of-Experts style architecture that attempts to combine a “temporal expert” (based on CycleNet) and a lightweight “cross-variable expert” (RCVTE) with a learned gating network and a Periodicity-Aware Local Curriculum Learning (PALCL) training protocol.

**Strengths:**

The high-level idea of combining separate modules that target temporal periodicity and cross-variable coupling is reasonable and aligns with prior modular/MoE thinking.

**Weaknesses:**

1.	The proposed system is largely an engineering assembly of existing pieces (CycleNet temporal module, a single-layer transformer for cross-variable interaction, a simple linear gating network, RevIN). The paper provides no compelling new modeling principle beyond “put these together and gate them.”
2.	The authors state broad superiority, but inspection shows the model achieves the best result on only four datasets/tasks
3.	Table 5 shows a limited ablation (only on six datasets) and omits a crucial ablation: the effect of PALCL itself. Given that PALCL is one of the principal methodological claims, the absence of an experiment that toggles PALCL on/off (with otherwise identical settings) is a serious oversight.
4.	The detailed architecture of Recurrent Cross-Variable Transformer Encoder (RCVTE) are not clear, the architected should be plotted.

**Questions:**

Provide an ablation that evaluates the model with and without PALCL
Replace RCVTE with simpler alternatives and report accuracy/computer.
Add parameter counts, FLOPs, and measured inference latency and VRAM usage to allow cost-benefit comparison.

---

> ### Author Response · Authors · 2025-11-26
> **Response for Weaknesses and Questions**
>
> Weakness:
>
> 1. Architectural Novelty: We have provided a comprehensive response regarding the novelty and architectural contributions of MoVE in Section 1, "Novelty and Architecture of MoVE," in our overall official comments. This section details the fundamental innovations of our approach.
>
> 2. Baseline Selection and Performance Consistency: Our evaluation focuses on TQNet, CycleNet, and iTransformer as primary baselines due to their architectural relevance and recency. While existing models demonstrate excellence on specific datasets (e.g., iTransformer on Traffic, TimeXer on ETTh2, CycleNet on ETTm2), they often exhibit inconsistent performance across other datasets, frequently with substantially higher parameter counts and computational overhead. In contrast, MoVE maintains robust performance across all datasets while preserving computational efficiency.
>
> 3. Enhanced Ablation Analysis: We have incorporated new ablation studies specifically examining the PALCL component in Table 5 and Section 4.5, providing detailed analysis of its individual contribution and synergistic effects within the overall architecture.
>
> 4. Architecture Clarification: As detailed in Section 1 of our official comments, the core innovation lies in our Recurrent Cross-Variable Transformer Encoder (RCVTE), which implements a single-layer cross-variable transformer with two key modifications: replacement of conventional attention with Robust Cross-Variable Attention (RCVA), and integration of Periodicity-Aware Local Curriculum Learning (PALCL) for stabilized training.
>
> Questions:
>
> 1.
>
> Performance Our method demonstrates consistent improvements over key baselines:
>
> Outperforms TQNet in 11/16 metrics
>
> Outperforms CycleNet in 13/16 metrics
>
> Outperforms iTransformer in 14/16 metrics
>
> Notably, Move's performance on Weather dataset has been updated using a daily cycle (period=6) and learning rate of 0.0001, with comprehensive results provided in Tables 3 and 4 due to the data and setting loss before .
>
> Efficiency Analysis: (Section 6.5, Table 7, 8)
>
> Average Parameter Counts: 1.53M
>
> Average FLOPS: 2089M
>
> Training Time: Lower bounded by the TQNet and Upper bounded by iTransformer

---

### Author Response · Authors · 2025-11-26
**Overall Response for Concerns of Novelty, Performance and Efficiency**

We sincerely thank for the PC, SAC and reviewers for their time and effort of reviewing this paper. Below we address the points raised regarding the novelty and performance of our work.

1. Novelty and Architecture of MoVE:

Section 3.2 is summarized below:

1.1 Recurrent Cross-Variable Transformer Encoder (RCVTE)

The RCVTE represents a novel architecture that fundamentally advances cross-variable relationship modeling in multivariate time series through two key innovations:

Recurrent Architecture with PALCL: Unlike standard transformers, RCVTE employs a recurrent design that processes both periodic components and raw inputs through the same encoder via our Periodicity-Aware Local Curriculum Learning (PALCL) strategy. This enables explicit modeling of periodic inter-channel relationships while maintaining parameter efficiency.

Robust Cross-Variable Attention (RCVA): We replace conventional attention with RCVA to address the limitation of standard softmax attention, which demonstrates susceptibility to attention collapse due to exponential function sensitivity to outliers

The PALCL strategy implements a structured two-phase curriculum within each batch, where RCVTE first processes noise-free periodic vectors to establish fundamental periodic correlations, then transitions to original data for complex pattern integration. This progressive approach ensures robust temporal interaction learning, as illustrated in Figure 1.

1.2 Gating Network

Apart from its basic functionality, our gating network introduces dynamic adaptation to address two critical challenges in multivariate time series:

Temporal Distribution Shift: The gate serves as an adaptive arbitrator between Channel Dependent (CD) and Channel Independent (CI) strategies, automatically tailoring the approach to each channel's specific distribution shift characteristics between training and testing data.

Model Collapse Prevention: Drawing parallels to recursive training collapse in generative AI (Shumailov et al., 2024), our gate prevents degeneration from input similarity by maintaining expert diversity and selectively prioritizing learning pathways.

This work presents, to our knowledge, the first approach that explicitly implements dynamic modulation across multiple experts to simultaneously mitigate distribution shift and prevent model collapse. This represents a significant departure from static architectures in existing cross-variable models including TimePro, TimeXer, and Crossformer. While SST incorporates some adaptive elements, its gating mechanism is primarily confined to balancing contributions between Patterns and Variations Experts, lacking the comprehensive multi-expert modulation demonstrated in our framework.

Key changes in Section 3.2:
To dynamically balance the contributions of the four experts, a lightweight gating network employs a softmax function. Apart from its basic functionality, This design strategically addresses two independent challenges in multivariate time series modeling.

First, it mitigates temporal distribution shift between training and test distributions, where severity varies across channels. As established in time series domain adaptation literature, such non-stationarity significantly degrades model performance. The gating mechanism serves as an adaptive arbitrator between Channel Dependent (CD) and Channel Independent (CI) strategies, favoring CI for channels with severe shift and CD for stable channels. This hybrid approach aligns with findings from \citep{CDCI}, enabling channel-specific resilience to temporal characteristics.

Second, the gating network prevents model collapse—a degenerative condition where neural networks fail when processing overly homogeneous or synthetic data patterns. Drawing parallels to the recursive training collapse observed in generative AI \citep{model_collpase}, PALCL strategy faces similar risks when learned periodic components substantially overlap with raw inputs. This input similarity can create degenerate learning signals, analogous to training on recursively generated data. The gating mechanism counters this by maintaining expert diversity, selectively prioritizing or discounting pathways to preserve learning signal variety.

The significance of the gating mechanism is evidenced by the substantial performance degradation observed upon its removal (Section 4.5.1, Table 5).


2. Performance
Our method demonstrates consistent improvements over key baselines:

Outperforms TQNet in 11/16 metrics

Outperforms CycleNet in 13/16 metrics

Outperforms iTransformer in 14/16 metrics

Notably, Move's performance on Weather dataset has been updated using a daily cycle (period=6) and learning rate of 0.0001, with comprehensive results provided in Tables 3 and 4 due to the data and setting loss before .

Efficiency Analysis: (Section 6.5, Table 7, 8)

Average Parameter Counts: 1.53M

Average FLOPS: 2089M

Training Time: Lower bounded by the TQNet and Upper bounded by iTransformer

---

### Meta-Review · Area_Chair_xYXs · 2026-01-07

**Summary:**

### Summary

The strengths of the work can be summarized as follows:
- The high-level motivation to decouple temporal modeling and cross-variable dependencies is considered clear, reasonable, and significant [a321, WPzk, A3WK].
- The introduction of Periodicity-Aware Local Curriculum Learning (PALCL) and Robust Cross-Variable Attention (RCVA) is recognized for improving training stability and alleviating attention collapse [goo3].
- The Mixture-of-Experts architecture is praised for offering better interpretability and adaptability compared to static fusion methods [goo3].

The weaknesses of the work can be summarized as follows:
- The proposed framework is criticized as an "engineering assembly" of existing pieces or too similar to prior works like SST, TimeXer, and TimePro [a321, A3WK, goo3].
- The model does not demonstrate broad superiority, achieving best results on only a limited number of datasets or metrics [a321, goo3].
- Reviewers found the visualization unappealing and unclear, and noted the manuscript content was insufficient or short [A3WK, goo3].
- The specific design of the Cross-Variable Expert (RCVTE) was found to be unclear or seemingly indistinguishable from simple cross-attention mechanisms [a321, WPzk].
- The initial submission omitted crucial ablations, such as the specific effect of the PALCL strategy [a321, A3WK, WPzk].

**Reviewer Concerns:**

**Reviewer a321**
- Addressed: The authors provided the requested ablation study for the Periodicity-Aware Local Curriculum Learning (PALCL) strategy and provided the requested efficiency metrics.
- Outstanding: The core concern that the system is merely an "engineering assembly"  remains a subjective disagreement. While the authors argued that the dynamic gating and RCVA are novel, the reviewer may still view the aggregation of CycleNet and Transformers as incremental.

**Reviewer WPzk**

- Addressed: The authors clarified the distinction between their approach and simple cross-attention by explaining the Robust Cross-Variable Attention (RCVA). They also addressed the "Periodic Dependency" concern by highlighting performance gains over CycleNet itself and provided an analysis of gating network weights.
- Outstanding: The reviewer might would still question whether the performance improvement primarily comes from the additional parameters introduced.

**Reviewer A3WK**

- Addressed: The authors stating that all important content has been included in the main text.
- Outstanding: The critique regarding Visualization remains largely outstanding; the authors defended the figure rather than promising a significant redesign. The concern that the method lacks innovation compared to TimeXer/TimePro was countered by the authors' claim of "dynamic adaptation", but this may not fully satisfy the reviewer's demand for distinct novelty.

**Reviewer goo3**

- Addressed: The authors provided a breakdown of win-rates against specific baselines to counter the "poor performance" claim.
- Outstanding: The similarity to the SST architecture remains a key concern. Although the authors highlight dynamic modulation as a difference, the overall structure still appears quite similar. The novelty is limited, and the model design and experimental support could be clearer and more convincing.

**Reviewer Scores:**

**Reviewer a321**: Likely to maintain score 2.

The authors successfully provided the missing data requested by this reviewer, specifically the efficiency metrics and the ablation study for PALCL. However, the reviewer's primary critique was fundamental: they viewed the system as an "engineering assembly" lacking "compelling new modeling principle" . Unless the reviewer changes their fundamental view on the novelty of combining these specific modules, simply adding missing experiments is unlikely to result in a significant score increase.

**Reviewer WPzk**: Likely to maintain score 4.

This reviewer focused on technical novelty and the necessity of key components such as the cross-variable modeling and curriculum learning strategy. Although the rebuttal provides clarifications and additional analyses, it does not fully address whether the performance gains, particularly those from PALCL, are due to genuine methodological improvements or increased model capacity. As a result, the response may prevent further concerns but is unlikely to change the original score.

**Reviewer A3WK**: Likely to maintain score 2.

This reviewer expressed strong negative opinions, rating the presentation as "poor" and stating the paper is "not good enough"  with high confidence. They criticized the visualization as "unappealing" and the innovation as limited compared to existing works like TimeXer . The authors' rebuttal defended the existing figure and page limits  rather than offering significant changes, which is unlikely to satisfy the reviewer's concerns about presentation and content sufficiency.

**Reviewer goo3**: Likely to maintain score 2.

This reviewer was very confident in their assessment that the innovation is limited, specifically citing a strong architectural similarity to the existing "SST" model. They also noted the model only achieved best results in less than half of the metrics. While the authors argued that their dynamic gating distinguishes them from SST , the lack of dominant performance across all metrics reinforces the reviewer's view that the method is an incremental improvement rather than a breakthrough.

---

### Decision · Program_Chairs · 2026-01-26

Reject